# Sustainability in Healthcare Sector: The Dental Aligners Case

Chiara Caelli [1,*], Francesco Tamburrino [2,*], Carlo Brondi [1], Armando Viviano Razionale [2],
Andrea Ballarino[1] and Sandro Barone [2]

1. CNR STIIMA—Institute of Intelligent Industrial Technologies and Systems for Advanced Manufacturing, National Research Council, Via Alfonso Corti 12, 20133 Milan, Italy; carlo.brondi@stiima.cnr.it (C.B.); andrea.ballarino@stiima.cnr.it (A.B.)
2. Department of Civil and Industrial Engineering, University of Pisa, Largo Lucio Lazzarino 1, 56122 Pisa, Italy; armando.viviano.razionale@unipi.it (A.V.R.); sandro.barone@unipi.it (S.B.)
* Correspondence: chiara.caelli@stiima.cnr.it (C.C.); francesco.tamburrino@unipi.it (F.T.)

**Abstract:** Additive manufacturing is a technology gaining ground in fields where a high degree of product customization is required; in particular, several aspects need to be explored concerning traditional technologies, such as the variety of materials and their consumption. It also remains to be clarified whether these technologies can contribute to the ecological transition when applied in healthcare. This study compares two technologies for producing clear dental aligners: thermoforming and direct 3D printing. The former method thermoforms a polymeric disc over 3D-printed, customized models. The second, more innovative approach involves directly printing aligners using Additive Manufacturing (AM), specifically applying Digital Light Processing (DLP) technology. The study conducts a comparative Life Cycle Assessment (LCA) analysis to assess the environmental impact of these two different manufacturing processes. The research results highlight that adopting direct printing through AM can bring advantages in terms of environmental sustainability, thanks to the reduction in raw materials and electricity consumption. These drops are drivers for the decreased potential environmental impacts across all impact categories considered within the EF 3.1 method. Furthermore, lowering the amount of raw material needed in the direct printing process contributes to a notable decrease in the overall volume of waste generated, emphasizing the environmental benefits of this technique.

**Keywords:** sustainability; healthcare; additive manufacturing; life cycle assessment

## 1. Introduction

Additive manufacturing is a processing technology that responds to high customization needs [1]. At the same time, its application can imply critical issues related to environmental aspects, such as the consumption of ad hoc materials and energy use; therefore, additive technology should be coupled with preliminary environmental assessment [2,3]. In such a perspective, there are specific sectors in which additive manufacturing could be profitable both in terms of functional performance and environmental efficiency: in fact, strong needs for customization characterize these sectors. A possible application for the healthcare sector is then presented.

Since the 20th century, the growing awareness of malocclusion and its related consequences have led to various orthodontic approaches for repositioning teeth. Adolescent and adult patients who are aware of their malocclusion traits and are not satisfied with their dental appearance tend to suffer from psychosocial concerns. However, the treatment of malocclusion has historically been related to invasive appliances that severely affect the wearer's daily life. This aspect has improved with the introduction of clear and removable aligners [4]. The standard manufacturing process of the aligners and the practice of companies producing an excess of orthodontic aligners are at odds with environmental concerns.

There is a growing demand from orthodontists to minimize the environmental impact of aligner production [5].

The present study aims to quantitatively analyze and compare two technologies for producing clear aligners from a sustainability standpoint. This comparison is possible through Life Cycle Assessment (LCA), a methodology regulated by ISO 14040:2006 [6] and ISO 14044:2006 [7]. LCA enables a quantitative assessment of the potential environmental impact associated with product manufacturing. The two analyzed technologies were thermoforming in-mould aligners and direct 3D printing. In particular, the Digital Light Processing (DLP) photopolymerization has been chosen among all AM technologies, with the primary data available. The innovative aspect of this study is the quantitative assessment of the life cycle impact for each of the two technologies analyzed and the following comparison regarding the same functional unit.

This study consists of four sections: Introduction, Materials and Methods, Results, and Conclusions and Future research. The former summarizes the state of the art and the evolution of technologies used in orthodontics. The second, aligned with reference standards, defines the goal and scope of the study and describes the necessary data inventory, focusing on materials, logistics, and energy. In the third section, the results and critical issues of the study are presented and discussed, and in the last paragraph, the study's conclusions are drawn.

*State of the Art*

Historically, in the early 1900s, malocclusion was treated by positioning metal rings cemented to teeth to support wires for applying moving forces. This approach caused many dental caries: it was almost impossible to maintain correct dental hygiene because of the limited offering of tools in the market, the mechanical encumbrance of the rings, and subsequent dental plaque formation [8]. The 1960s saw the introduction of stainless steel brackets to support wires, while in the 1970s, transparent or translucent non-metallic brackets were used.

In 1997, a significant development occurred in orthodontics with the introduction of the first aligners designed for orthodontic treatment. This innovation was driven by patient demand for more comfortable and less intrusive methods to improve dental alignment. Unlike traditional braces, aligners are aesthetically pleasing and do not cause discomfort to the lips. Additionally, their transparent appearance and the ability to remove them while eating and maintaining oral hygiene are additional benefits [8,9]. However, clear aligner therapy involves altering the teeth's position, angulation, and rotation, thereby enhancing all parameters necessary for proper and healthy occlusion and articulation. While it may address mild non-extraction cases more quickly and efficiently, it does demand more time compared to fixed appliance treatment for patients with more complex issues [4,9]. It is important to emphasise that it makes sense to compare different technologies for producing aligners capable of acting on the same problems and not between other orthodontic treatment techniques.

In the past decade, the most widespread technology for producing clear aligners was the manufacture of dental models that express the desired tooth movement. The traditional process involves five steps: acquiring the original dental anatomy, developing the teeth movement model, 3D printing of models, thermoforming, and the cutting of the aligners [10,11]. The thermoforming equipment is shown in Figure 1.

Three-dimensional printing, also called additive manufacturing, is a set of technologies based on adding materials, usually layer upon layer, through different techniques (material extrusion, material jetting, powder bed fusion, vat photo-polymerization, etc.), and this production process might be impractical for large-scale production. Still, it can be especially applicable for all those applications requiring a high level of individual customization, such as dental aligners [10–12]. Recent techniques have enabled clinicians to print the aligners directly, eliminating model production and thermoforming steps. The direct additive manufacturing technique increases efficiency and reduces waste, eliminating inaccuracies

associated with the 3D printing of models and thermoforming processes. Moreover, several advantages over conventional fabrication are present: digitally designed borders, smooth edges, no need for undercuts, higher precision without errors introduced by model moulding and thermoforming, and customizable intra-aligner thickness. However, in the current context, there is a low diffusion of this methodology of manufacturing aligners, mainly related to some limitations of the technologies and materials involved [8,10,11]. This study involves a sustainability-oriented comparison of these two technologies for producing clear aligners. It could catalyze aligner companies to invest in research and development in the additive manufacturing approach.

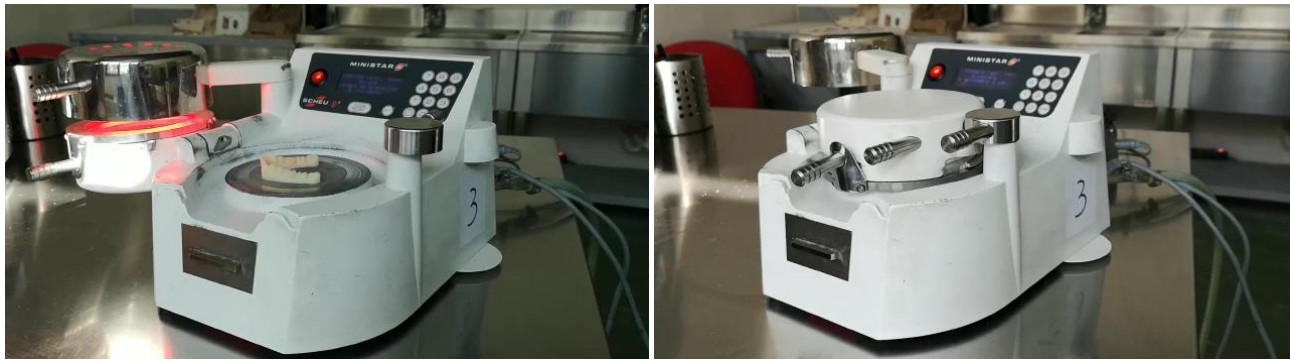

**Figure 1.** Thermoforming equipment.

## 2. Materials and Methods

The direct 3D printing of aligners is a technology with many plus points, but being less established as a process, it has some limiting factors. The choice of material to be used is one of them. Indeed, the material should be biocompatible and transparent, with low stiffness and good elasticity, resilient and resistant to its use in human saliva [4]. Moreover, these materials have to be selected among the classified as Class IIa long-term biocompatible resins and conform to the essential requirements and provisions of the Council Directive 93/42/EEC and Medical Device Directive 2007/47/EC [13].

The biocompatibility of aligners is one of the most discussed issues because patient safety has to be assured. When dealing with additive manufacturing of photosensitive resins (e.g., through technologies like Material Jetting and VAT photopolymerization), the toxicity of 3D printed materials decreases as they undergo post-polymerization. Therefore, post-curing is essential to eliminate the toxicity levels, removing uncured resin and making the printed material safer for intraoral usage [14]. The recommended protocol suggests UV curing and subsequent washing; these steps guarantee increased mechanical properties and reduced cytotoxicity [15]. With traditional technology, different polymers are used to produce the 3D-printed model and the thermoforming of aligners. The material used for the model comprises acrylic monomers and oligomers, and the supports are made of a mixture of acrylic oligomers, glycols and glycerine. Differently, PETG copolyester is the chosen material for thermoforming the aligners. Additive manufactured aligners can reduce the variety of materials used as it is no longer necessary to have a model on which the plastic disc is thermoformed. Various additive manufacturing technologies might be used to print clear aligners directly: fused deposition modelling, selective laser sintering, selective laser melting, direct pellets fused deposition, stereolithography, multi-jet photo-cured polymer process, or continuous liquid interface production technology [11,16].

However, VAT photopolymerization (VPP) is currently the most appropriate choice among AM technologies due to its exceptional resolution and accuracy. Additionally, its ability to achieve high transparency in 3D printed components makes it a viable option and is, in part, already used in orthodontics. Moreover, in the case of Digital Light Processing (DLP) technology, the material used for the support is the same as the one used for the aligners. This could contribute to the goal of reducing material variety. DLP is part of the VAT photopolymerization family, in which a liquid photopolymer is put in a resin tank

and is selectively cured by light-activated polymerization to create a solid polymer. In particular, in DLP-based approaches, light exposure is carried out by projecting a single digital image (mask), polymerizing every single layer simultaneously, reducing the printing time compared to other technologies [8,11].

This study consists of a comparative life cycle analysis between two processes for producing dental aligners, one more familiar and one innovative. The software OpenLCA v1.11.0 and the database Ecoinvent v3.9.1 were used to perform the numerical calculations. An attributional approach was adopted for the study, with the choice of database processes built according to the allocation cut-off method.

## 2.1. Goal and Scope Definition

In the International Standard ISO 14040 [6] are indicated the mandatory information that should be declared: the goal and the scope of the study, the functional unit and the target audience. The purpose of this study is to quantify the environmental burdens associated with the manufacturing of dental aligners to allow a comparison of the potential impacts related to the two production technologies. The chosen functional unit is 40 dental aligners, which constitute an average complete set needed for treatment, with their packaging. Each pair of aligners is used for two to three weeks and, after this time, replaced with the next one for continued treatment. The analyzed system includes the production of precursors from raw materials, logistics, and the aligners' manufacturing: this is a cradle-to-gate study. Since primary data on the use and disposal of aligners are unavailable, it is impossible to include the use and disposal steps. In the production processes, many steps are considered, from the cast production to the finishing and packaging, and the difference between the materials used is considerable. In fact, the polymer used for thermoforming can not be used for direct 3D printing of the aligner. The system boundaries are reported in Figures 2 and 3 for the thermoforming and the additive manufacturing techniques, respectively.

The target audience of this study consists of healthcare professionals, especially dentists and orthodontists, researchers, LCA practitioners, and additive manufacturing operators. In almost all cases, primary data, measured directly in the field, were used for all technologies analyzed. Support was received from AirNivol Srl, an Italian company specializing in designing and producing dental aligners, which provided a substantial portion of its primary production data. However, some modelling was necessary for the polymers present in the process and was carried out starting from the scientific literature and the procedures already available in the Ecoinvent database.

## 2.2. Inventory Characterization

As a cradle-to-gate analysis, the starting point involves extracting raw materials and producing precursors and materials used in manufacturing. This phase is similar for all three technologies, differing only in the materials used. The data used are primary in quantities, but processes already present in the database were used for modelling. Since no chemicals were available, ad hoc processes were created in some cases, exploiting the technical information in the literature, such as the most widespread synthesis method, and the process flows already present within Ecoinvent. The table with all the chosen database files is reported in Appendix A.

The first steps of manufacturing are the production of the plaster cast and its scanning; these are not always performed in the system considered because, in about 80% of cases, a digital scan is provided by the dentist, and therefore, it is not necessary to produce the cast starting from the impression. Thus, a mass allocation was performed in the modelling: only 20% of the material and energy consumption related to these two steps was counted. From this point forward, thermoforming and 3D printing technology modelling differ significantly and will be described individually in the following subsections.

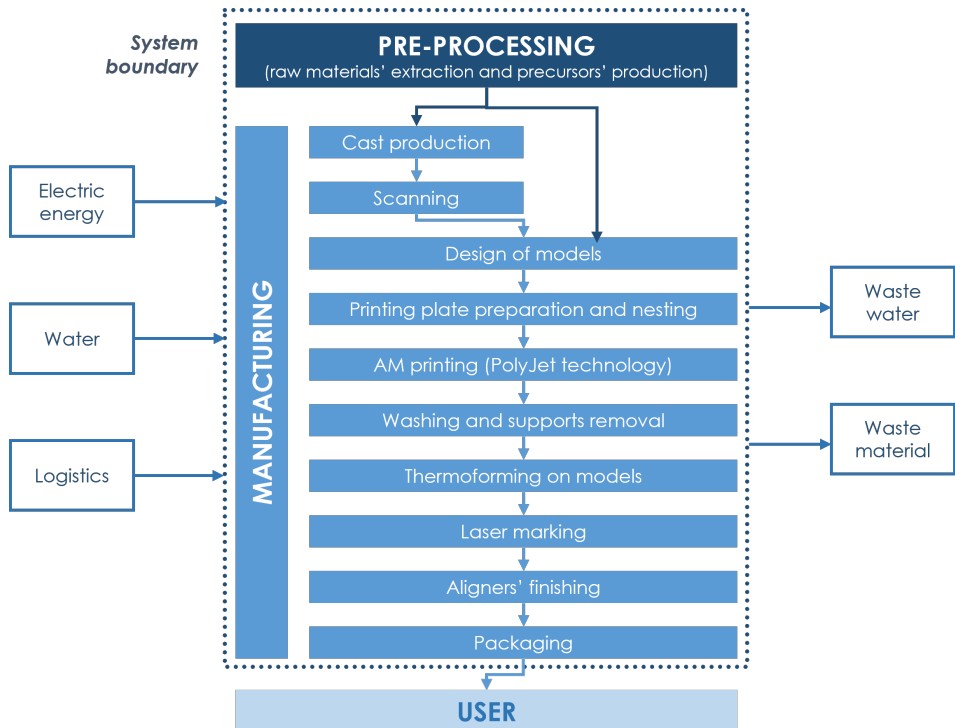

**Figure 2.** System boundary of the process which uses the thermoforming technology.

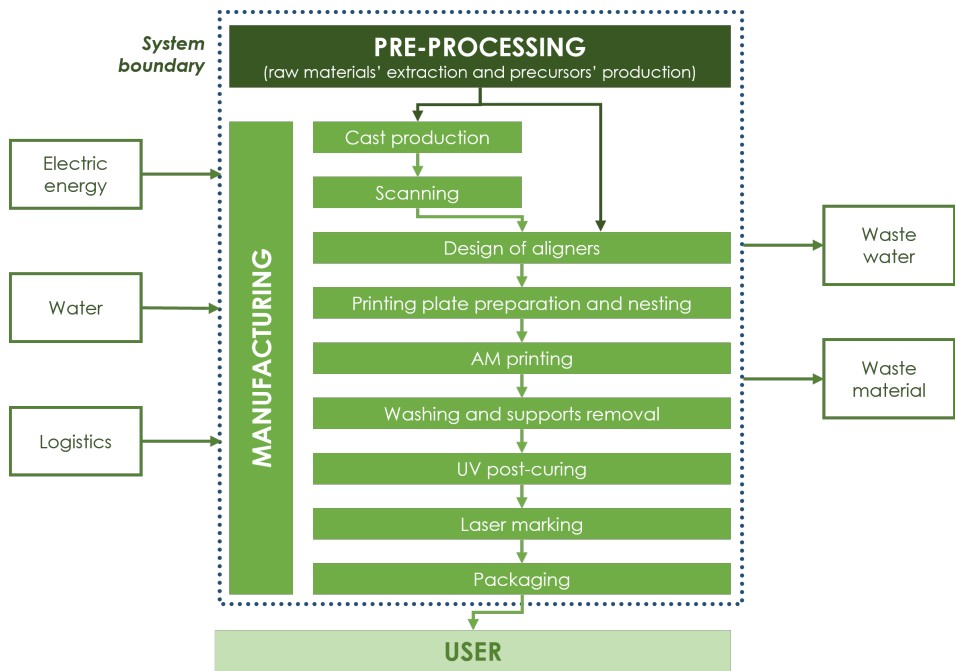

**Figure 3.** System boundary of the process using additive manufacturing technology.

### 2.2.1. Thermoforming

Once the scan of the patient's dental arches has been obtained, technicians proceed to the models' design on which plastic discs will be thermoformed. The following steps are preparing the printer, cleaning its plate and nesting, and 3D printing the models required for the entire treatment with PolyJet technology. This technology belongs to the Material Jetting additive manufacturing method. It uses multiple print heads to deposit ultra-thin layers of liquid material onto a build platform, which are cured instantly with UV light. It is renowned for producing high-resolution, multi-material, and full-color 3D-printed objects.

Two different polymeric materials are needed to print the models: one for the models and one for the supports. A fraction of these materials are considered to be discarded during the purging process of the 3D printer. After printing, the supports are removed by washing with high-pressure water jets. The resulting wastewater, contaminated with the printing material, is then sent to the water treatment plant. Following the cleaning of the models, thermoforming is performed: a disc of thermoplastic material, specifically a polyethylene terephthalate glycol (PETG) copolymer, is heated and moulded over the model to obtain the dental aligner. The result of the thermoforming application is shown in Figure 4.

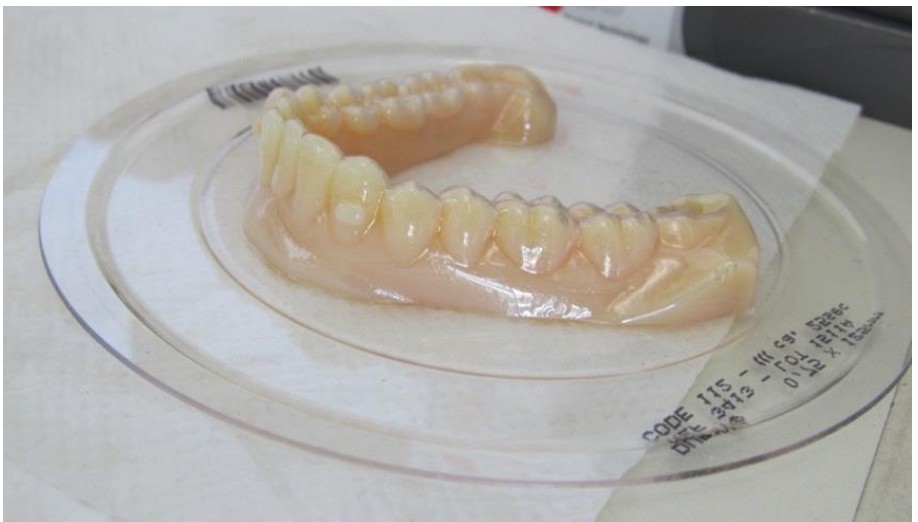

**Figure 4.** Results of the thermoforming process: aligner thermoformed on the printed model.

The following stages are laser marking, the cutting of aligners, where excess material is removed with a milling machine, and packing. The packaging includes plastic bags in which the aligners are inserted in pairs, a rigid plastic case, a cardboard box, and the information material.

### 2.2.2. Additive Manufacturing

The exact starting point is assumed to be the same as the thermoforming procedure for direct printing. With the scan of the patient's dental arches, the design of the aligners can begin. Then, after computer modelling, the printer can be prepared and nested, and the aligners' direct printing can occur. The primary distinction between this technology and thermoforming lies in the absence of printed models. In this case, the aligners are printed directly, reducing waste and material usage and eliminating inaccuracies associated with both the 3D printing of models and the thermoforming process. The printing result is shown in Figure 5. The DLP is part of the VPP family and consists of selective curing of a liquid photopolymer: the light exposure is performed by projecting a single mask to polymerize every single layer simultaneously [8,11].

The composition of the material used for the direct printing of aligners is described by [17] as an aliphatic vinyl ester-urethane polymer, possibly cross-linked with methacrylate functionalization.

After printing, the supports necessary for this step are removed manually, followed by a 30 min water wash, to eliminate residuals of resin that were not polymerized and part of the remaining supports. Then, UV post-curing is essential in producing dental aligners because it guarantees increased mechanical properties and reduced cytotoxicity of the materials used. In the present study, a UV post-curing duration of 30 min was considered. After laser marking, the dental aligners are packed using plastic bags, in which the pairs of aligners are inserted; the rigid plastic case and the information material are added to the cardboard box of the user package.

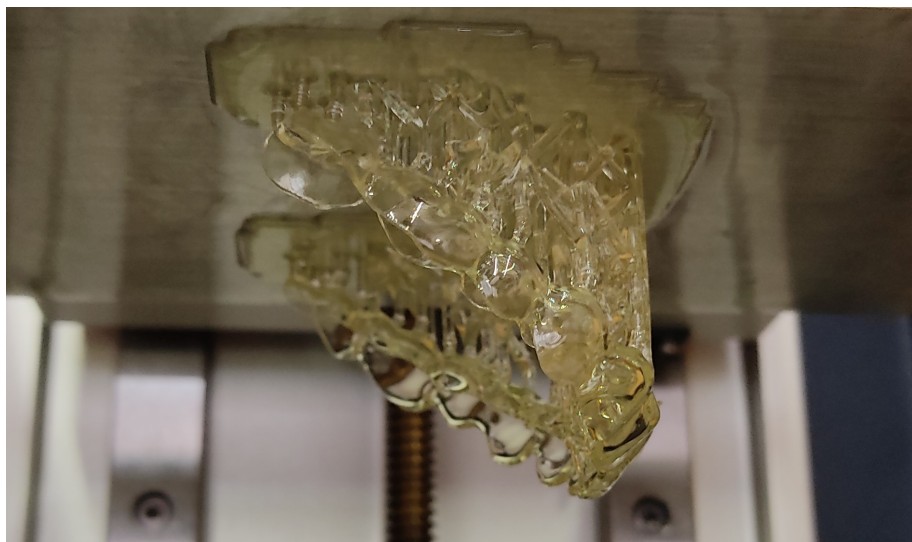

**Figure 5.** Result of direct printing DLP technology: the clear aligner with supports.

### 2.2.3. Materials

The modelling of the polymeric materials, used both for the production of the model, in the case of thermoforming, and for the production of the aligners, was carried out from the existing sheets in the Ecoinvent database. As the materials used were not available in the database, the precise composition of which was obtained from the safety material data sheets, modelling of the most common synthesis routes was used, taking information from the literature regarding the operating conditions and energy required for production. Following this method, an attempt was made to remain as faithful as possible to the materials used, avoiding generic data sheets. In the case of plastic packaging materials, such as bags and rigid cases for aligners, the approach involved consulting the data sheets from the specific plastic material database. Additionally, the manufacturing processes, bag extrusion and injection moulding for rigid cases were integrated into the analysis to provide a comprehensive understanding.

### 2.2.4. Energy and Logistics

Concerning modelling the electricity used, the only energy source in the system boundary considered is the Italian residual mix ("electricity, low voltage, residual mix | electricity, low voltage | Cutoff, U (IT)"), as it is taken from the national grid because there are no certificates on the energy purchased. The system boundary chosen also includes the transport phase of all the raw materials to the manufacturing step. For most materials, data sheets have been selected from the market-type database, mainly at a European level, which is already considered an average transport, as more detailed information is unavailable. Instead, as regards the transport of the packaging, for which the distances are known, expressed in km, but not the specific means of transportation, the data sheet "market for transport, freight, lorry, unspecified | transport, freight, lorry, unspecified | Cutoff, U (RER)" was chosen.

## 3. Results

In this section, the life cycle assessment results are presented. They include an examination of the primary drivers for each technology discussed and a comparative analysis between thermoforming and the considered additive manufacturing application. Moreover, the hypotheses and the sensitivity analysis results will be discussed. The method chosen to assess potential environmental impacts is EF 3.1, which provides results on 25 impact categories. The results are presented for the main categories, and there will be a particular focus on four of them; even in the field of additive manufacturing, different impact assessment methods and different categories are used, and by following what is reported

in the literature, it was chosen to delve deeper into these categories: climate change, energy resources, acidification, and eutrophication [18,19].

### 3.1. Comparative Analysis

According to ISO 14044 [7], in a comparative study, the systems are evaluated with the same functional unit and equivalent methodological considerations, such as performance, system boundary, and data quality. The chosen functional unit is a complete set of 40 aligners capable of performing orthodontic treatment on several cycles. This study is cradle-to-gate, i.e., it starts with the extraction of raw materials, their processing to obtain intermediate products, the actual production of the aligners by thermoforming or direct printing, and their packaging. Thus, the chosen functional unit is the same for both technologies, and the system boundaries are perfectly comparable.

The results of this comparative study, referring to a functional unit, are shown in Figure 6 in terms of potential environmental impacts for the main impact categories of the EF 3.1 method. The numerical values on which the plots are based are reported in Appendix B. When analyzing the overall results over the whole system, it can be observed that thermoforming has a higher potential impact for all impact categories considered concerning direct 3D printing. The smallest gap occurs in the "Land use" category with a 31% change, and the more significant gap occurs in the "Ozone depletion" category with a 98% reduction in impact. An in-depth analysis focused on the main vectors between the categories of materials, wastes, energy, transport, and packaging was carried out to investigate the reasons that led to such clear-cut results in more detail.

As mentioned above, the impact categories selected are Acidification, Climate change, Energy resources and Eutrophication, freshwater, marine and terrestrial, and the detailed results are shown in Figures 7 and 8.

Analyzing in detail the comparison of the impacts represented in Figures 7 and 8, it is observable that for thermoforming, the main impact is given for almost all impact categories by the raw materials used. Another significant portion of the impact is provided by energy and waste, depending on the category considered. It is also worth noting that there is a significant decrease in the impacts of direct printing using DLP technology, attributable to the material, energy, and waste contributions. Conversely, since the packaging is the same for the two technologies, there is an impact with the same absolute value but a much greater relative value in the case of 3D printing.

### 3.2. Interpretation and Discussion

The consistent reduction in environmental impact, for all impact categories considered, for additive technology compared to thermoforming is substantially due to the decrease in the quantity of material and energy used, depicted in Figure 9. In fact, by using direct printing technology, there is no longer the consumption of material and electricity linked to printing the model; therefore, waste material and water use will also be lower. Furthermore, the significant difference in the quantity of water used is due to the washing of the soluble supports after printing the models in the case of thermoforming, using special equipment; on the contrary, for direct printing, only a small amount of water is needed to remove the residuals of resin that were not polymerized.

### 3.3. Limitations of the Study

The study's main limitation is linked to the materials used for producing the models, thermoforming and direct printing. The material safety data sheets, when available, report only the families to which the mixture's components belong and not the specific compound. Within the particular case of the material used for direct printing, not even the safety data sheet is available as it has a patented composition; therefore, the modelling of such material is based on information available in the literature. Furthermore, only limited chemical compounds are available in the Ecoinvent database, and a simulation of a realistic synthesis route and its operating conditions is necessary.

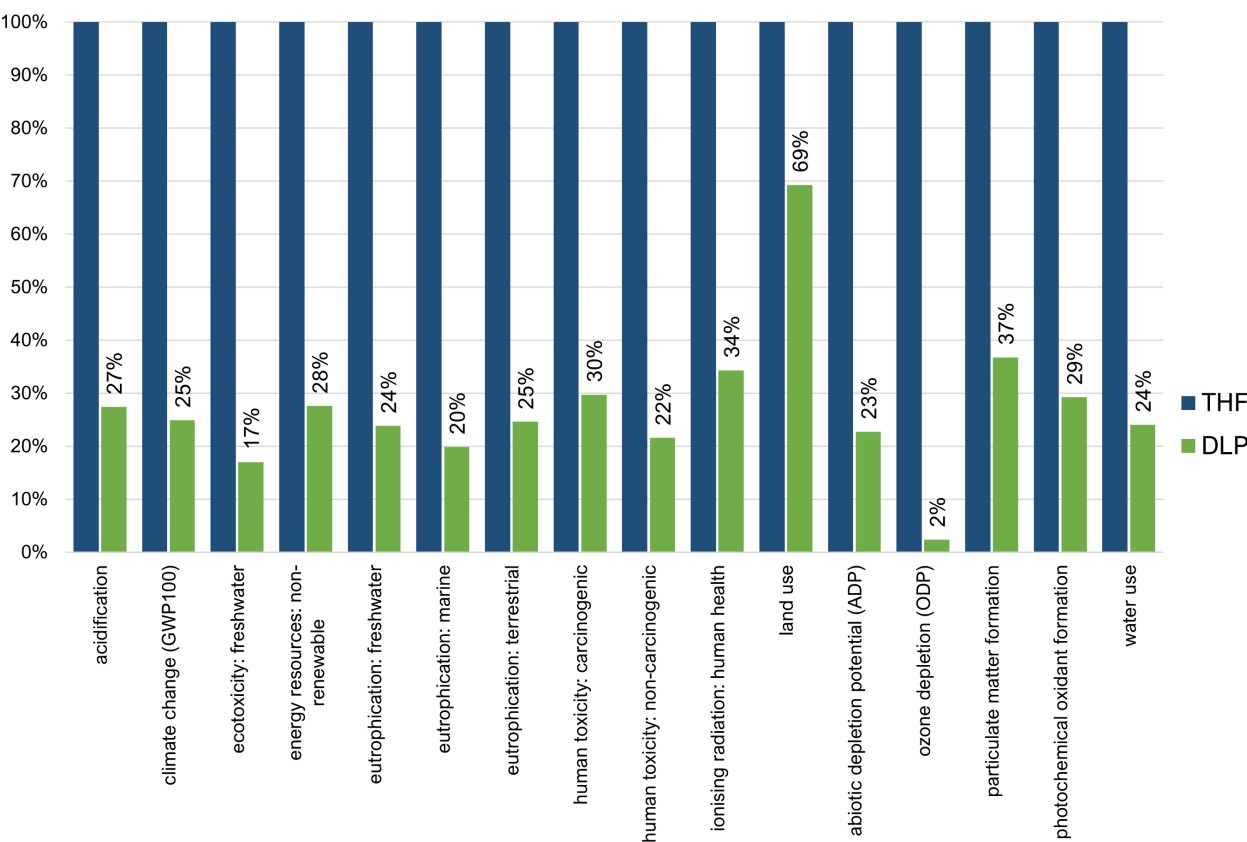

**Figure 6.** Results of the comparative LCA calculation performed with EF 3.1 impact method.

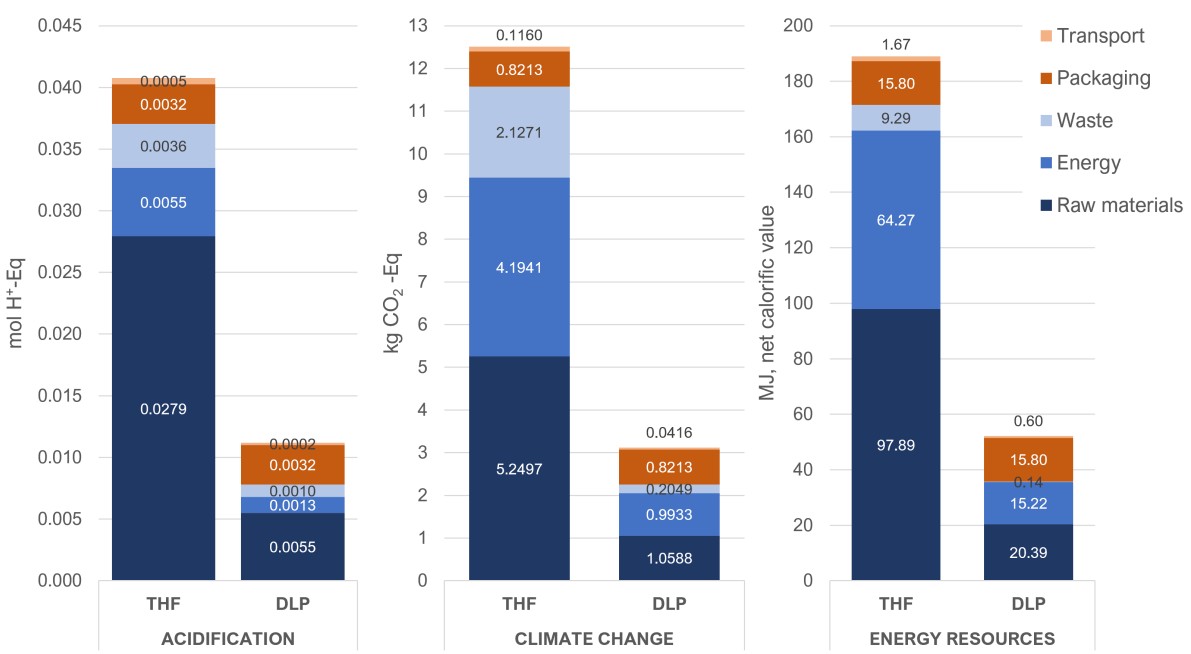

**Figure 7.** Impacts per FU for acidification, climate change and energy resources impact categories.

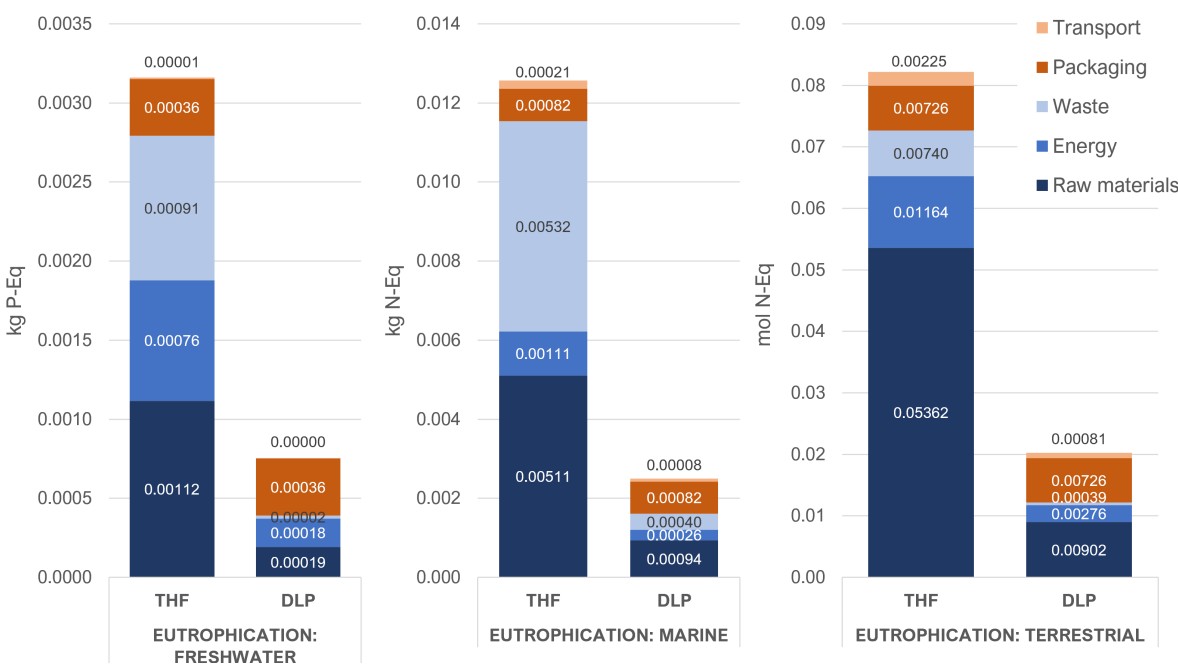

**Figure 8.** Impacts per FU for the eutrophication impact category.

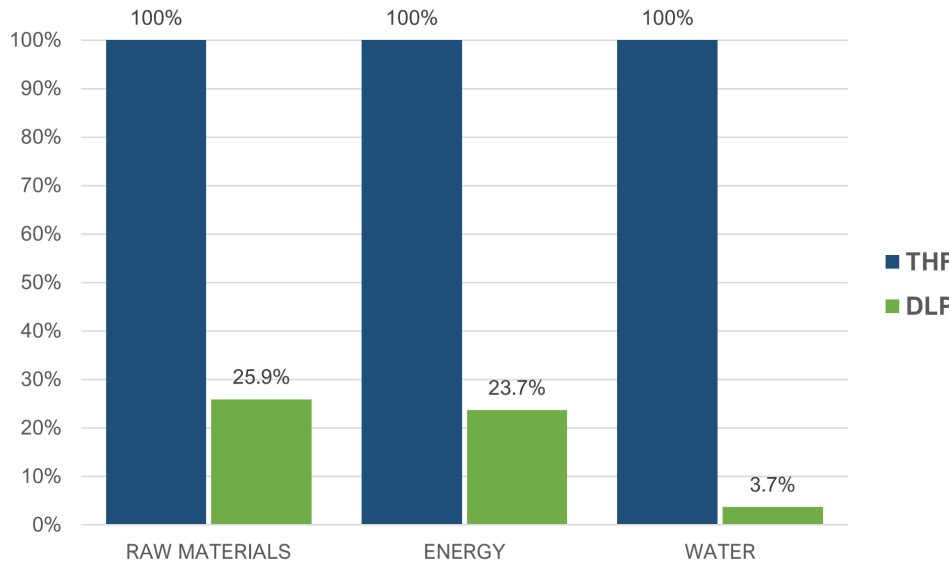

**Figure 9.** Percentage reduction in energy and material consumption between technologies.

## 4. Conclusions and Future Research

In this comparative LCA study, the environmental impacts associated with two production processes of dental aligners have been evaluated: thermoforming and direct 3D printing through Digital Light Processing (DLP). This analysis aimed to provide insights into the sustainability of these processes considering a cradle-to-gate perspective, encompassing all life cycle stages, from raw material extraction to aligner fabrication. The new methodology demonstrates a substantial reduction in environmental impacts by decreasing the amount of material used. It is necessary to contextualise the results, mentioning that this technology was analyzed for the specific application of dental aligners and, therefore, the application of environmental effectiveness in the biomedical field should be extended to other possible areas of application, comparing it with different technologies and for long-lasting items.

It should also be mentioned that the study refers to an exploratory application in the biomedical field. The large-scale dissemination and actual environmental impact should also include market factors related to costs and raw material availability. Only qualitative considerations can be made from a cost perspective regarding the two presented methods. From a live production cost standpoint, direct printing of aligners may be less competitive than thermoforming. This difference is primarily due to a significant impact on the final cost of the printing material, given the low availability of compatible materials, certified for dental use, and possessing the appropriate mechanical properties. However, it is essential to highlight that direct printing avoids the fixed costs associated with producing models for thermoforming and their disposal. In a future perspective, where the number of available materials could increase, prices would likely be reduced. An extended assessment that considers environmental and economic aspects could explore future perspectives for making informed decisions in the dental aligner production industry, aligning with the broader goals of sustainability and responsible manufacturing practices. A final aspect concerns the improvability of the presented model, which should include an update of the existing databases and a greater disclosure of the material composition, overcoming patent barriers.

In conclusion, the results of this study suggest that the direct printing of dental aligners has a lower environmental impact than thermoforming. This advantage is primarily attributed to the efficient use of materials, reduced energy consumption, and minimized waste generation associated with DLP.

**Author Contributions:** Conceptualization, F.T. and C.B.; Methodology, C.C. and C.B.; Investigation, C.C., F.T. and A.V.R.; Software, C.C.; Formal analysis, C.C.; Data curation, F.T. and A.V.R.; Visualization, C.C.; Writing—original draft, C.C., F.T. and C.B.; Writing— review & editing, C.C., F.T., C.B. and A.V.R.; Supervision, A.B. and S.B.; Project administration, A.B. and S.B. All authors have read and agreed to the published version of the manuscript.

**Funding:** This research received no external funding.

**Data Availability Statement:** The data presented in this study are available on request from the corresponding author.

**Conflicts of Interest:** The authors declare no conflict of interest.

## Appendix A

The table that includes all of the references for the Ecoinvent 3.9.1 database sheets used to represent the two distinct production processes can be found below.

**Table A1.** Sheets selected from the Ecoinvent 3.9.1 database.

| Tech | Process Step | Item | Provider |
|---|---|---|---|
| THF | Cast production | Cast material | market for gypsum, mineral ǀ gypsum, mineral ǀ Cutoff, U (RER) |
| THF | Cast production | Cast material | market for transport, freight, lorry, unspecified ǀtransport, freight, lorry, unspecified ǀ Cutoff, U (RER) |
| THF | Cast production | Water | market for tap water ǀ tap water ǀ Cutoff, U (Europe without Switzerland) |
| THF | Cast production | Cast | market for waste gypsum ǀ waste gypsum ǀ Cutoff, U (Europe without Switzerland) |
| THF | Scanning | Electric energy | electricity, low voltage, residual mix ǀ electricity, low voltage ǀ Cutoff, U (IT) |
| THF | Design of models | Electric energy | electricity, low voltage, residual mix ǀ electricity, low voltage ǀ Cutoff, U (IT) |
| THF | Printing plate preparation and nesting | Electric energy | electricity, low voltage, residual mix ǀ electricity, low voltage ǀ Cutoff, U (IT) |
| THF | AM printing | Electric energy | electricity, low voltage, residual mix ǀ electricity, low voltage ǀ Cutoff, U (IT) |
| THF | AM printing | Model material | Model material |
| THF | AM printing | Model material | market for transport, freight, lorry, unspecified ǀtransport, freight, lorry, unspecified ǀ Cutoff, U (RER) |
| THF | AM printing | Support material | Support material |
| THF | AM printing | Support material | market for transport, freight, lorry, unspecified ǀtransport, freight, lorry, unspecified ǀ Cutoff, U (RER) |

**Table A1.** *Cont.*

| Tech | Process Step | Item | Provider |
|------|--------------|------|----------|
| THF | AM printing | Purges | market for hazardous waste, for incineration ∣ hazardous waste, for incineration ∣ Cutoff, U (Europe without Switzerland) |
| THF | Supports removal | Electric energy | electricity, low voltage, residual mix ∣ electricity, low voltage ∣ Cutoff, U (IT) |
| THF | Supports removal | Water | market for tap water ∣ tap water ∣ Cutoff, U (Europe without Switzerland) |
| THF | Supports removal | Wastewater | market for wastewater, average ∣ wastewater, average ∣ Cutoff, U (Europe without Switzerland) |
| THF | Thermoforming | Aligners material | PETG |
| THF | Thermoforming | Electric energy | electricity, low voltage, residual mix ∣ electricity, low voltage ∣ Cutoff, U (IT) |
| THF | Laser marking | Electric energy | electricity, low voltage, residual mix ∣ electricity, low voltage ∣ Cutoff, U (IT) |
| THF | Finishing | Electric energy | electricity, low voltage, residual mix ∣ electricity, low voltage ∣ Cutoff, U (IT) |
| THF | Finishing | Material waste | market for waste plastic, mixture ∣ waste plastic, mixture ∣ Cutoff, U (IT) |
| THF | Packaging | Plastic bags | LDPE bag |
| THF | Packaging | Cardboard box | folding boxboard carton production ∣ folding boxboard carton ∣ Cutoff, U (RER) |
| THF | Packaging | Info book | paper production, woodcontaining, supercalendered ∣ paper, woodcontaining, supercalendered ∣ Cutoff, U (RER) |
| THF | Packaging | Byte case | PP case |
| THF | Packaging | Plastic bags | market for transport, freight, lorry, unspecified ∣ transport, freight, lorry, unspecified ∣ Cutoff, U (RER) |
| THF | Packaging | Cardboard box | market for transport, freight, lorry, unspecified ∣ transport, freight, lorry, unspecified ∣ Cutoff, U (RER) |
| THF | Packaging | Info book | market for transport, freight, lorry, unspecified ∣ transport, freight, lorry, unspecified ∣ Cutoff, U (RER) |
| THF | Packaging | Byte case | market for transport, freight, lorry, unspecified ∣ transport, freight, lorry, unspecified ∣ Cutoff, U (RER) |
| AM DLP | Cast production | Cast material | market for gypsum, mineral ∣ gypsum, mineral ∣ Cutoff, U (RER) |
| AM DLP | Cast production | Cast material | market for transport, freight, lorry, unspecified ∣ transport, freight, lorry, unspecified ∣ Cutoff, U (RER) |
| AM DLP | Cast production | Water | market for tap water ∣ tap water ∣ Cutoff, U (Europe without Switzerland) |
| AM DLP | Cast production | Cast waste | market for waste gypsum ∣ waste gypsum ∣ Cutoff, U (Europe without Switzerland) |
| AM DLP | Scanning | Electric energy | electricity, low voltage, residual mix ∣ electricity, low voltage ∣ Cutoff, U (IT) |
| AM DLP | Design of aligners | Electric energy | electricity, low voltage, residual mix ∣ electricity, low voltage ∣ Cutoff, U (IT) |
| AM DLP | Printing plate preparation and nesting | Electric energy | electricity, low voltage, residual mix ∣ electricity, low voltage ∣ Cutoff, U (IT) |
| AM DLP | AM printing | Electric energy | electricity, low voltage, residual mix ∣ electricity, low voltage ∣ Cutoff, U (IT) |
| AM DLP | AM printing | Aligner material | Aligner AM DLP |
| AM DLP | AM printing | Aligner material | market for transport, freight, lorry, unspecified ∣ transport, freight, lorry, unspecified ∣ Cutoff, U (RER) |
| AM DLP | AM printing | Material waste | market for waste plastic, mixture ∣ waste plastic, mixture ∣ Cutoff, U (IT) |
| AM DLP | AM printing | Electric energy | electricity, low voltage, residual mix ∣ electricity, low voltage ∣ Cutoff, U (IT) |
| AM DLP | Supports removal | Water | market for tap water ∣ tap water ∣ Cutoff, U (Europe without Switzerland) |
| AM DLP | Supports removal | Wastewater | market for wastewater, average ∣ wastewater, average ∣ Cutoff, U (Europe without Switzerland) |
| AM DLP | UV post-curing | Electric energy | electricity, low voltage, residual mix ∣ electricity, low voltage ∣ Cutoff, U (IT) |
| AM DLP | Laser marking | Electric energy | electricity, low voltage, residual mix ∣ electricity, low voltage ∣ Cutoff, U (IT) |
| AM DLP | Packaging | Plastic bags | LDPE bag |
| AM DLP | Packaging | Cardboard box | folding boxboard carton production ∣ folding boxboard carton ∣ Cutoff, U (RER) |
| AM DLP | Packaging | Info book | paper production, woodcontaining, supercalendered ∣ paper, woodcontaining, supercalendered ∣ Cutoff, U (RER) |
| AM DLP | Packaging | Byte case | PP case |
| AM DLP | Packaging | Plastic bags | market for transport, freight, lorry, unspecified ∣ transport, freight, lorry, unspecified ∣ Cutoff, U (RER) |
| AM DLP | Packaging | Cardboard box | market for transport, freight, lorry, unspecified ∣ transport, freight, lorry, unspecified ∣ Cutoff, U (RER) |
| AM DLP | Packaging | Info book | market for transport, freight, lorry, unspecified ∣ transport, freight, lorry, unspecified ∣ Cutoff, U (RER) |
| AM DLP | Packaging | Byte case | market for transport, freight, lorry, unspecified ∣ transport, freight, lorry, unspecified ∣ Cutoff, U (RER) |

## Appendix B

The comparative Life Cycle Assessment study's numerical results, including all of the impact categories typical of the EF 3.1 method, are shown in the table below.

**Table A2.** Numerical results of the comparative analysis reported by impact category.

| Impact Category | THF | AM DLP |
|-----------------|-----|--------|
| acidification | $4.077 \times 10^{-2}$ | $1.119 \times 10^{-2}$ |
| climate change (GWP100) | $1.251 \times 10^{1}$ | $3.120 \times 10^{0}$ |
| climate change: biogenic | $4.602 \times 10^{-2}$ | $1.537 \times 10^{-2}$ |
| climate change: fossil | $1.245 \times 10^{1}$ | $3.099 \times 10^{0}$ |

**Table A2.** *Cont.*

| Impact Category | THF | AM DLP |
|---|---|---|
| climate change: land use and land use change | $9.599 \times 10^{-3}$ | $5.742 \times 10^{-3}$ |
| ecotoxicity: freshwater | $1.309 \times 10^{2}$ | $2.228 \times 10^{1}$ |
| ecotoxicity: freshwater, inorganics | $9.837 \times 10^{1}$ | $1.264 \times 10^{1}$ |
| ecotoxicity: freshwater, organics | $3.257 \times 10^{1}$ | $9.637 \times 10^{0}$ |
| energy resources: non-renewable | $1.889 \times 10^{2}$ | $5.215 \times 10^{1}$ |
| eutrophication: freshwater | $3.160 \times 10^{-3}$ | $7.543 \times 10^{-4}$ |
| eutrophication: marine | $1.257 \times 10^{-2}$ | $2.500 \times 10^{-3}$ |
| eutrophication: terrestrial | $8.218 \times 10^{-2}$ | $2.023 \times 10^{-2}$ |
| human toxicity: carcinogenic | $6.141 \times 10^{-9}$ | $1.827 \times 10^{-9}$ |
| human toxicity: carcinogenic, inorganics | $3.350 \times 10^{-9}$ | $6.145 \times 10^{10}$ |
| human toxicity: carcinogenic, organics | $2.791 \times 10^{-9}$ | $1.212 \times 10^{-9}$ |
| human toxicity: non-carcinogenic | $1.462 \times 10^{-7}$ | $3.158 \times 10^{-8}$ |
| human toxicity: non-carcinogenic, inorganics | $1.359 \times 10^{-7}$ | $2.975 \times 10^{-8}$ |
| human toxicity: non-carcinogenic, organics | $1.030 \times 10^{-8}$ | $1.830 \times 10^{-9}$ |
| ionising radiation: human health | $9.190 \times 10^{-1}$ | $3.155 \times 10^{-1}$ |
| land use | $7.305 \times 10^{1}$ | $5.065 \times 10^{1}$ |
| abiotic depletion potential (ADP) | $8.306 \times 10^{-5}$ | $1.889 \times 10^{-5}$ |
| ozone depletion (ODP) | $3.798 \times 10^{-6}$ | $9.062 \times 10^{-8}$ |
| particulate matter formation | $4.518 \times 10^{-7}$ | $1.662 \times 10^{-7}$ |
| photochemical oxidant formation | $3.041 \times 10^{-2}$ | $8.898 \times 10^{-3}$ |
| water use | $4.322 \times 10^{0}$ | $1.041 \times 10^{0}$ |

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
