# Peer review of "Sustainability in Healthcare Sector: The Dental Aligners Case"

_sustainability, doi:10.3390/su152416757_

Round 1

Reviewer 1 Report

Comments and Suggestions for Authors

This manuscript takes the dental aligner as an example to study the sustainability in the healthcare sector. A comparative life cycle assessment (LCA) analysis was used to assess the environmental impact of thermoforming and 3D printing process. The subject of this work is interesting. I recommend to accept it for publication after minor revisions.

1.     It is recommended to supplement the sources and basis of the various data used in Figures 6-9.

2.     It is recommended that the authors discuss and exchange ideas with experts in the fields of 3D printing and thermoforming to gain a deeper understanding of the details in these two manufacturing processes, in order to further improve the LCA evaluation in this study.

Comments on the Quality of English Language

 I recommend to accept it for publication after minor revisions.

Reviewer 2 Report

Comments and Suggestions for Authors

The manuscript is supposed to compare between thermoforming and 3D printing of dental aligners from a sustainability perspective.  The paper needs significant improvement.

1) The language of the paper is hard to read. It is full of grammatical mistakes and awkward statements. The writing sounds like a casual conversation between people rather than a scientific document.

2) the organization of the paper is not easy to follow.

3) the manuscript is full of redundant and unrelated information. 

4) The paper doesn't clearly show how the results were obtained.  

Comments on the Quality of English Language

The language of the paper is hard to read. It is full of grammatical mistakes and awkward statements. The writing sounds like a casual conversation between people rather than a scientific document.

Reviewer 3 Report

Comments and Suggestions for Authors

The paper (sustainability-2711747 entitled ‘’Sustainability in healthcare sector: the dental aligners case’’) conducted a comparative life cycle assessment (LCA) analysis to assess the environmental impact of two different manufacturing processes used to produce transparent dental 2 aligners. The study methods are up-to-date valid and reliable, and data represented properly. The conclusions seem to have answered the aims of the study. The paper has potential to be published.

However, there are some points that needs to be taken into consideration as follows. Please:

·         Abstract is well-written and brief. However, adding just one or two introductory sentences would be beneficial. Sentences might be general sentences including AM, healthcare sector and sustainability to explain why you have preferred to investigate this topic in the present paper. For instance, sentences can be about regarding AM applications in healthcare sector and the importance of healthcare sector and sustainability, the main reason of conducting LCA. Then you had better be mentioning what was covered in the paper. The abstract in this form looks like the method section.

·         Line 27: please first give the long version of terms then give the abbreviation e.g. Life Cycle Assessment (LCA). Correct it through the whole paper.

·         Paragraph in academic writing should include min 250 max 500 words ideally. Please combine paragraphs in section 1.0 making it a paragraph. If you press enter in the end of each sentence it makes the text look like watsap conversation. Paragraphs can be added only when the topic has been changed. Please correct it through the whole paper.

·         Line 32, add 1-3 sentence to give the structure of the paper, i.e. the paper is constructed into 5 sections. Section 1 covers…, finally the conclusion section includes …

·         Line 30, use past tense here ‘are thermoforming’ as you are mentioning what was achieved/used in the past.

·         Line 55 nowadays is a very vague word. Never use it in any academic text. Instead use e.g. in the past decade, .. please be always specific.

·         Line 76 use ‘consists of’ and omit will.

·         Line 197, please check if the abbreviation is VP or VPP. For me it should be VP.

·         Line 244, use results are presented, omit will.

·         Figure 8, remove the vertical line on the left-hand side. Numbers are not quite readable as their text size is small. Please make text size completable with figure 9.

·         Please expand Section 4 by adding more evaluative comments and conclusion.

Comments on the Quality of English Language

The paper is well written however the use of grammar and tenses should be checked

Reviewer 4 Report

Comments and Suggestions for Authors

Please find the reviewer's comment as an attachment.

Round 2

Reviewer 2 Report

Comments and Suggestions for Authors

The manuscript has undergone significant improvements. However, it could benefit from an additional round of editing, as several sentences remain awkward to read.

Comments on the Quality of English Language

The manuscript could benefit from an additional round of editing, as several sentences remain awkward to read

Author Response

Further editing of the paper was carried out, correcting syntax errors and making the text more readable and fluent.